# Amputation for Extremity Sarcoma: Indications and Outcomes

**DOI:** 10.3390/cancers13205125

**Published:** 2021-10-13

**Authors:** Maya Kirilova, Alexander Klein, Lars H. Lindner, Silke Nachbichler, Thomas Knösel, Christof Birkenmaier, Andrea Baur-Melnyk, Hans Roland Dürr

**Affiliations:** 1Musculoskeletal Oncology, Department of Orthopaedics and Trauma Surgery, University Hospital, Ludwig-Maximilians-University Munich, D-81377 Munich, Germany; m.kirilova@medius-kliniken.de (M.K.); alexander.klein@med.uni-muenchen.de (A.K.); christof.birkenmaier@med.uni-muenchen.de (C.B.); 2SarKUM, Center of Bone and Soft Tissue Tumors, University Hospital, Ludwig-Maximilians-University Munich, D-81377 Munich, Germany; Lars.Lindner@med.uni-muenchen.de (L.H.L.); silke.nachbichler@med.uni-muenchen.de (S.N.); Thomas.Knoesel@med.uni-muenchen.de (T.K.); Andrea.Baur@med.uni-muenchen.de (A.B.-M.); 3Department of Medicine III, University Hospital, Ludwig-Maximilians-University Munich, D-81377 Munich, Germany; 4Department of Radiation Oncology, University Hospital, Ludwig-Maximilians-University Munich, D-81377 Munich, Germany; 5Institute of Pathology, University Hospital, Ludwig-Maximilians-University Munich, D-81377 Munich, Germany; 6Department of Radiology, University Hospital, Ludwig-Maximilians-University Munich, D-81377 Munich, Germany

**Keywords:** sarcoma, surgery, amputation, prognosis, local recurrence, survival

## Abstract

**Simple Summary:**

Sarcomas are malignant tumors of soft tissues or bone. While limb salvage surgery (LSS) is the standard treatment, amputation is an option especially in local recurrence (LR) or complications after LSS. Two groups with primary amputations (*n* = 120) or secondary amputations after failed LSS due to LR or complications (*n* = 29) were compared. Five-year LR-free survival was 84% and 17 (16%) patients developed LR, of which 16 were in group I and only one in group II. Overall survival (OS) at five years was 44%, and the rate was identical in both groups. In those group II patients who had a secondary amputation after LSS due to contaminated margins or LR (*n* = 12) five-year OS was 33% compared to 48% in patients with complications (*n* = 17). This study indicates the worse oncological outcomes with respect to OS of sarcoma patients needing an amputation as compared to LSS. Patients with primary amputation or those who had a secondary amputation after failed LSS for whatever reason showed the same results.

**Abstract:**

Background: Sarcomas are rare, malignant tumors of soft tissues or bone. Limb salvage surgery (LSS) is the standard treatment, but amputation is still an option, especially in local recurrence or complications after LSS. Methods: We retrospectively reviewed indications and oncological outcomes in patients who underwent an amputation. Two groups with either primary amputations (*n* = 120) or with secondary amputations after failed LSS with local recurrence or complications (*n* = 29) were compared with the main end points of LRFS and OS. Results: Five-year LRFS was 84% with 17 (16%) patients developing local recurrence, of which 16 (13%) occurred in group I. Forty-two (28%) patients developed metastatic disease and overall survival at five years was 44%. Overall survival (OS) was the same in both groups. In those group II patients who had a secondary amputation due to LR or insufficient margins after LSS (*n* = 12) the five-year OS was 33% compared to 48% in patients with amputation due to complications (*n* = 17) (n.s.). Conclusions: This study indicates the worse oncological outcomes with respect to OS of sarcoma patients requiring an amputation as compared to LSS. Patients with primary amputation or those who had a secondary amputation after failed LSS for whatever reason showed the same oncological results.

## 1. Introduction

Sarcomas are rare, malignant tumors of soft tissues or bone with an incidence of about 2 per 100,000 inhabitants and a predilection for the lower extremities [1,2,3,4,5]. In a 1982 randomized trial comparing limb salvage surgery (LSS) with radiation therapy (RT) to amputation, no benefit for the latter was obvious [6]. Limb salvage surgery has since become the standard treatment in extremity sarcoma surgery [7].

Despite the advances in LSS, including free vascular flaps or extended neurovascular resections and reconstructions, amputation is still a valid option. If limb function is insufficient, local recurrence (LR) with widespread contamination leaves no other option. If infection and/or ischemia after LSS could not be treated otherwise, amputation is still indicated [8]. In the rare cases with exulcerating, fungating tumors, amputation might be the most appropriate palliative procedure.

There are studies in osteosarcoma patients which describe a better local control with amputation but no survival benefit over LSS in patients with intralesional or marginal margins [9] but also two meta analyses showing higher five-year survival rates for LSS [10,11]. Regarding soft tissue sarcomas, no difference in overall survival could be shown in two studies [12,13]. Regarding primary or secondary amputations in localized extremity sarcoma, no difference in oncological outcome was published by Erstad et al. in 2018 [14,15].

We therefore retrospectively reviewed our experience in respect to indications and oncological outcomes in patients with extremity sarcoma who underwent an amputation between 1980 and 2018. Two groups of patients with either primary or secondary amputations after failed LSS with local recurrence or complications were compared: we sought to investigate the question, of whether patients who undergo an amputation due to local complications might have a better prognosis than those who require an amputation because of LR or for contaminated margins after a LSS.

## 2. Patients and Methods

After approval by our Institutional Review Board, we retrospectively reviewed 149 sarcoma patients who had undergone amputation at the authors’ institution between 1980 and 2018. Patients with prior limb salvage surgery (LSS) at other institutions were also included, and several patients had received chemotherapy and radiotherapy, as is stated in Table 1.

We divided the patients into two groups with either primary amputation (Group I) (Figure 1) or secondary amputation after inadequate margins (Figure 2) or local recurrence in LSS or following any type of complication such as infection or ischemia (Figure 3).

For local staging, magnetic resonance imaging (MRI) and in some cases computed tomography (CT) were used to clarify the location of the tumor and the extension in respect to vital structures such as vessels, nerves or neighboring compartments. A CT scan of the chest or in early years a thoraxic radiograph was used for diagnosing pulmonary metastatic disease. This and also local MRI was repeated for follow-up. In all resections the margin status was evaluated by using the method of applying ink to the specimen.

With exception of those patients that required amputations for non-tumor associated complications of LSS, all patients had been discussed at an interdisciplinary tumor board at our institution prior to surgery.

The margin was defined according to the FNCLCC-grading system as being R0 if a layer of healthy tissue around the lesion was present (wide resection) or as R1 if the margins were contaminated but the tumor capsule remained closed (marginal resection). In very few patients that were already in a palliative procedure, amputations with tumor left at the resection margins were performed. This situation was classified as an R2 resection. 

### Endpoints and Statistics

In this retrospective study, follow-up of the patients was analyzed in respect to local recurrence (LR) and metastatic disease with the main end points being LRFS and OS. All patients were followed for evidence of LR or distant metastasis as described above. LRFS and OS were defined either as the time from amputation to the first occurrence of LR or to death from any cause. For statistical analysis, OS and LRFS were calculated according to the Kaplan-Meier method. Significance analysis was performed using the Log-Rank or the Cox Proportional-Hazards Regression model. A *p* value of less than 0.05 was considered statistically significant. The data analysis software used was MedCalc^®^ (MedCalc Software, Ostend, Belgium).

## 3. Results

### 3.1. Patient Characteristics

The median age of the 92 male and 57 female patients was 58 years (mean 54, range 13–89). Only seven children, (13–17 years) all with bone sarcomas, had been included. The median tumor size was 10 cm (mean 11, range 1–25). Forty (27%) patients had metastatic disease at the time of diagnosis (Table 1 and Table 2).

120 patients (81%) had a primary (group I), and 29 (19%) a secondary amputation (group II). As is typical in amputations, the resection margins were only rarely contaminated (6%) in both groups. Fifty-eight (39%) underwent chemotherapy either before or/and after surgery, whereas radiotherapy was used in 13% of patients. 

The indications for amputation for both groups are listed in Table 1. In group I, multicompartmental involvement and the size of the tumor were the main reasons for amputation. Despite multicompartmental involvement, LSS is possible in many cases. But in those patients who had amputations for that indication in this study reconstruction of the bone, the vessels, the nerves or the soft tissues in total adds up to an individually unacceptable surgical risk or functional disadvantage in respect to amputation. In group II, infection or ischemia after LSS were the major factors leading to amputation. Local recurrence (LR) as cause of a secondary amputation was seen in 14% of this group. The lower extremity comprised more than two thirds of all amputations with transfemoral amputations in about a third of all patients (Table 2).

24 (16%) patients had a follow-up less than 12 months after surgery. 61 (41%) patients deceased during follow-up. 

### 3.2. Metastatic Disease, Local Recurrence-Free Survival and Overall Survival

Five-year LRFS was 84%, and 17 (11%) patients developed local recurrence whereas 42 (28%) patients developed metastatic disease after surgery within a median of four months (0–120). Overall survival of the entire cohort at five years was 44%.

### 3.3. Comparison of Both Groups

Overall survival (OS) was identical in both groups (Figure 4), and as a function of the high number of patients with metastatic disease, comparably low. In group II, only one patient developed LR more than two years after amputation. In group I, 13% of the patients developed LR. Due to the low number of patients at risk this difference did not reach significance (Figure 5). When LR did occur, OS was markedly reduced (Figure 6), but also without reaching statistical significance. In those patients with LR as the indication for amputation, OS was worse, but again, without statistical significance (Figure 7).

In Group II, those patients having a secondary amputation due to LR or insufficient margins after LSS (*n* = 12), five-year OS was 33% compared to 48% in patients with secondary amputation due to complications (*n* = 17) (n.s.).

## 4. Discussion

In this study, patients with bone and soft tissue sarcomas, including eight patients who needed an amputation at the level of the pelvis, were included. As stated above, amputation for oncological reasons may be considered a bias in respect to worse oncological outcome. Vascular infiltration is a known worse prognostic factor in osteosarcoma as also bone invasion is in soft tissue sarcoma [16,17,18]. The involvement of neurovascular structures in comparison has either no influence or a less significant influence on prognosis [17,18]. Also, larger size, which in many cases together with the infiltration of neurovascular structures predicts amputation, is a well-established single worse prognostic factor [19].

A separation of entities and locations may have advantages because we know that both factors do influence treatment and prognosis of the patients. But at the end such a small number of patients in the subgroups would result that drawing any conclusions would be difficult. We examined that issue in the literature. Papakonstantinou et al. published 2020 a meta analysis of osteosarcoma patients only treated either by LSS or amputation. The numbers of amputated patients in those studies were: 53, 27, 38, 40, 42, 36, 15, 300, 15, 95, 48, 46 and 143. In total 9/13 studies had a number below 50 patients. The studies with larger numbers, such as 143 or 300 are out of nationwide cohorts such as SEER or the Japanese register [11]. Those register studies, of course, allow large numbers in precisely defined subgroups such as pelvic chondrosarcoma patients with a profound matching of 131 patients in each of two groups (amputated vs. LSS, National Cancers Database, Chicago, IL, USA) [20] but they share all the disadvantages of retrospective nationwide databases. More than those national registers, meta analyses of data as for osteosarcoma only (all age) (934 LSS vs. 662 amputated) might attract high numbers but would also present difficulties in comparing the single studies [10]. Single center studies do have problems in reaching sufficient numbers. A large study of amputated patients published from Brigham and Women’s and Dana-Faber in 2018 had 54 extremity sarcoma (including “buttock”) patients of mixed bone and soft tissue sarcomas in 10 years [15]. There are some studies including only subgroups as soft tissue sarcomas but they ended with small numbers such as 18 [13] or 39 [21]. Even mixed groups of bone and soft tissue sarcomas from recent years reached sometimes only small numbers, such as 24 [22]. If the authors try to focus on location as distal tibia and entity as osteosarcoma, the resulting numbers are as small as 19 amputations even in a large center such as the Rizzoli [12], or 25 patients with soft tissue sarcoma of the extremities at Mount Sinai Hospital, Toronto [23]. Very few studies end with sufficient monocentric numbers in clearly defined subgroups, such as the 2015 published study from Birmingham comparing 197 patients with LSS to 127 amputated patients in extremity osteosarcomas only [9]. Furthermore, Rizzoli published their osteosarcoma only data (location “limb”) with 95 amputations in 2002 [24].

In our group of sarcoma patients, an amputation had to be performed in about 10% of cases and these data parallel the experience of other institutions [21]. In general, patients with a need for amputation do have a worse prognosis since they typically have larger tumors, involvement of critical structures or multicompartmental local recurrences [9,23,25]. Comparing our own, recently published data regarding OS in deep seated soft tissue sarcomas [26] with those of this current study, five-year OS was 75% in G2 sarcomas compared to 66% and 64% in G3 sarcomas compared to 31%, respectively, in the current study. So the need to amputate is bad news for these patients also in terms of their overall prognosis. This assessment is also strengthened by a recent study showing an almost twofold increase in five-year overall survival in patients with osteosarcoma who had LSS as compared to those with amputation [11].

The major causes leading to primary amputation were the involvement of multiple compartments and the size of the tumor in critical locations, which is consistent with the literature [23,25,27]. In secondary amputations, contaminated margins or LR which did not allow for an appropriate wide resection with another LSS counted for 41% of the cases. 59% of the patients had a failure of LSS, especially an infection, which constitutes a well-known issue. In a long-term follow-up study by Grimer et al, the risk of amputation was 16% at 30 years in patients with endoprosthetic replacement for malignant tumors of bone [28].

Our hypothesis that those patients who had the secondary amputation due to local complications (and not a tumor related issue) might have a better prognosis than those with a secondary amputation due to LR or contaminated margins could not be proven on the basis of statistical significance. However, a trend towards such a difference was apparent, and with only 29 patients in group II (versus 120 in group I), this could potentially also be caused by a lack of statistical power.

Patients with primary and secondary amputations did have the same prognosis (Figure 4). This finding is identical to the results published by Stevenson et al. [21]. In their small series of 39 patients, those with primary or secondary amputations showed nearly the same five-year OS as in our study. Stevenson et al. argue that the prognosis of the amputees is worse as compared to the literature in STS in general. We could prove that by comparison with our own published data of the total cohort as stated above [26]. Also, Mavrogenis et al. in their study of osteosarcoma patients at the distal tibia did not see any differences regarding survival or LR [12]. In the total group of 465 LSS and 95 amputations in osteosarcomas of the limb published from the Rizzoli Institute in 2002, the same finding was evident [24].

Local recurrence was evident in only one patient (3%) in Group II but in 16 (13%) in Group I. We think that this represents a bias because 59% of the patients in Group II had an amputation due to a non-tumor related complication of LSS. Stevenson et al. also observed 13% of LR in their series [21]. As LR in general in STS is in the same range [26], this finding is astonishing. One would assume that LR is reduced after amputation as compared to LSS. We think this might be the effect of selection bias in this very specific group of patients. The main reason for the worse OS was metastatic disease in both group of patients with also those patients with non-tumor related complications forcing amputation showing a considerable rate of metastatic disease.

In summary, amputation is still a valid option in treating sarcoma patients. Patients who had undergone primary amputation due to tumor location and extent had the same prognosis as patients secondarily amputated for complications of LSS, tumor-associated or not. The prognosis of amputated patients proved to be worse in comparison to published data of sarcoma resections in general. LR was seen as often as in LSS. The high numbers of metastatic disease reflect the selection bias of this group of patients. For clinical practice, a secondary amputation after failed LSS does therefore not influence the oncological outcome of the patient but might influence the amputation level.

## 5. Limitations of the Study

This is a retrospective study covering a period of 38 years. The diagnostic and therapeutic options for sarcoma patients have changed considerably during this long period of time, but the principles of limb sparing surgery have remained the same over the study period. Functional considerations and results had not been investigated, but of course influenced the indication for the procedures. The study cohort consists of bone and soft tissue sarcoma patients in different locations. A separation of entities and locations may have advantages, but the general aspects of surgical sarcoma therapy apply to all. We are well aware that this study does not investigate or consider the known prognostic factors in sarcoma patients. This study cohort of amputees is highly selected in respect to worse prognostic factors in the group of patients amputated for oncological reasons.

## 6. Conclusions

This study demonstrates worse oncological outcomes in respect to the overall survival of sarcoma patients that require an amputation as opposed to those patients qualifying for limb-sparing surgery. Patients with primary amputations had the same oncological results as those who had an amputation after failed LSS for any reason.

## Figures and Tables

**Figure 1 cancers-13-05125-f001:**
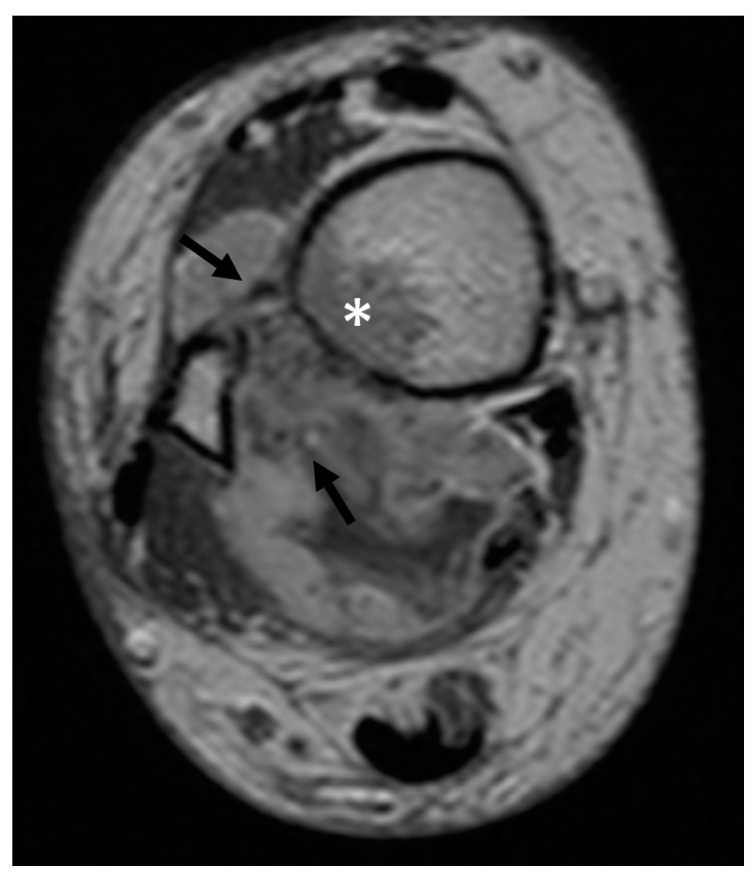
Axial MRI of an Undifferentiated Pleomorphic Sarcoma of the right lower leg infiltrating the bone (*) and the major vessels and nerves (→).

**Figure 2 cancers-13-05125-f002:**
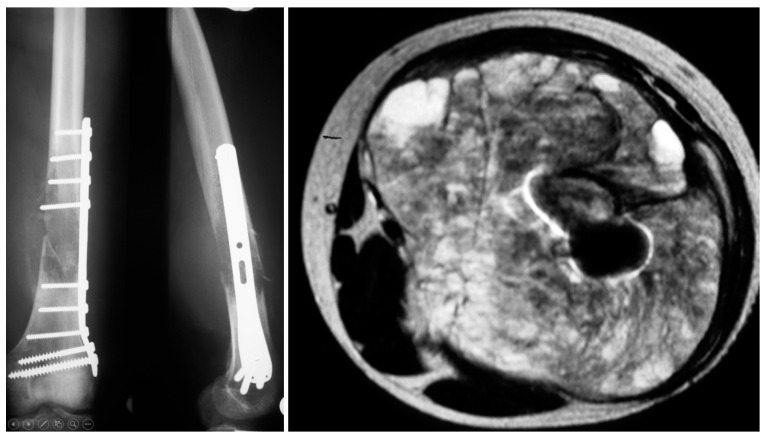
Radiographs and axial MRI of a patient with an osteosarcoma. After pathological fracture an osteosynthesis has induced widespread contamination and tumor growth of the whole distal upper calf.

**Figure 3 cancers-13-05125-f003:**
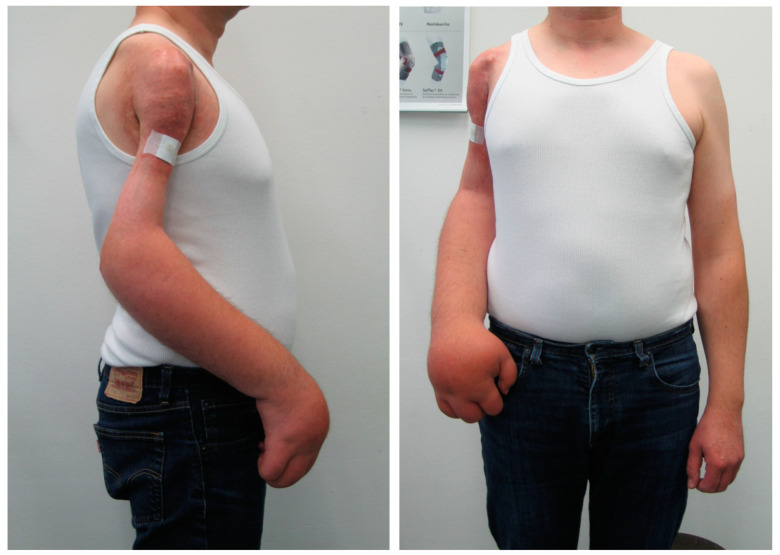
More than 30 years after treatment of a Ewing sarcoma of the humerus with local radiation and chemotherapy complications such as chronic osteomyelitis, nerve palsy and edema, an amputation due to functional reasons was indicated.

**Figure 4 cancers-13-05125-f004:**
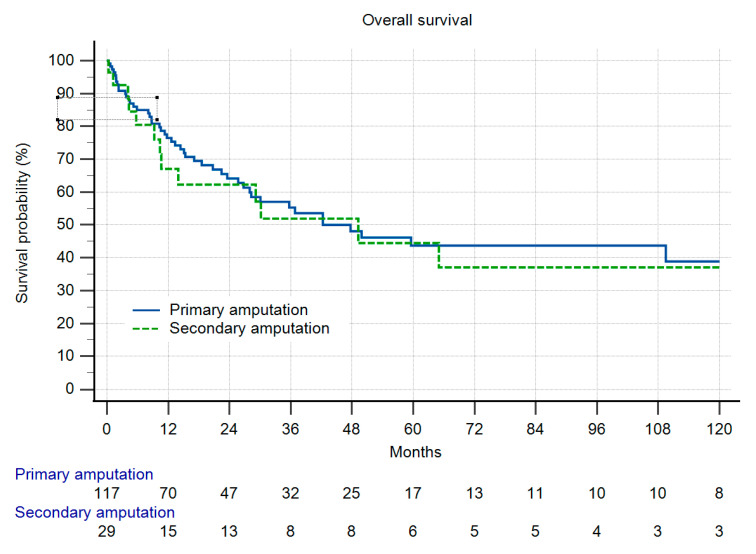
Overall survival (OS) in 146 patients with primary or secondary amputations (three patients excluded due to insufficient data), n.s.

**Figure 5 cancers-13-05125-f005:**
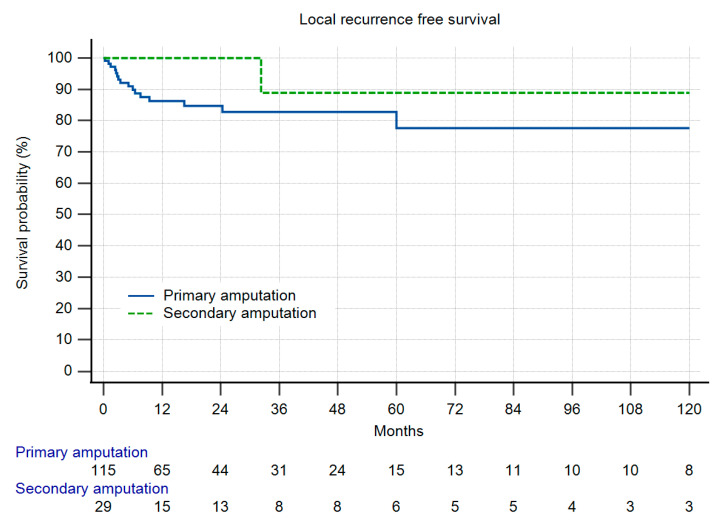
Local recurrence-free survival (LRFS) in 144 patients with primary or secondary amputations (five patients excluded due to insufficient data), n.s.

**Figure 6 cancers-13-05125-f006:**
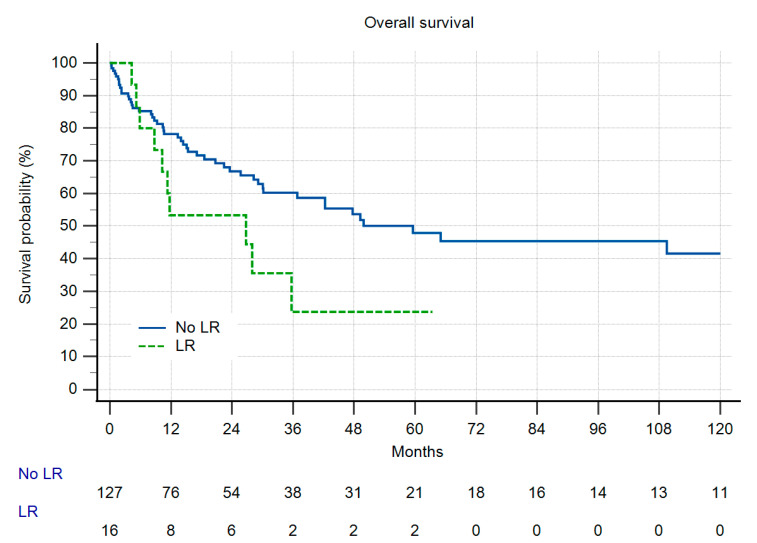
Overall survival by local recurrence after amputation, *n* = 143, six patients excluded due to insufficient data, n.s. (*p* = 0.0642).

**Figure 7 cancers-13-05125-f007:**
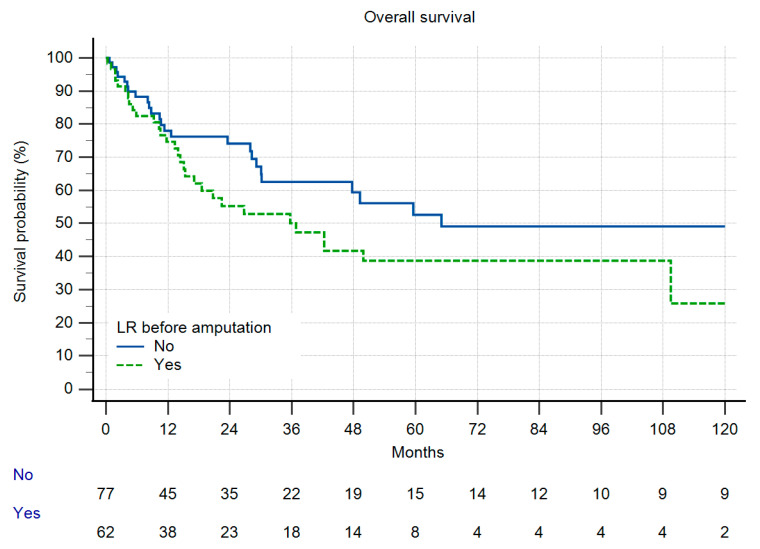
Overall survival by local recurrence before amputation, *n* = 139, 10 patients excluded due to insufficient data, n.s. (*p* = 0.0625).

**Table 1 cancers-13-05125-t001:** Indications, metastatic disease, adjuvant therapies and outcomes data. Percentage in brackets.

	Total (*n* = 149)	Group I (*n* = 120)	Group II (*n* = 29)	*p* Value
Indication for primary amputation				
Multiple compartments involved		65 (56%)		
Size		31 (27%)		
Neurovascular involvement		11 (9%)		
Bone involvement		7 (6%)		
Combined		3 (3%)		
Indication for secondary amputation				
Local recurrence			4 (14%)	
Contaminated margins after LSS			8 (28%)	
Infection or ischemia after LSS			17 (59%)	
Metastatic disease				
Before amputation	40 (27%)	28 (23%)	12 (41%)	n.s.
After amputation	42 (28%)	35 (29%)	7 (24%)	n.s.
Margins				
R0	140 (94%)	114 (95%)	26 (90%)	
R1	6 (4%)	5 (4%)	1 (3%)	
R2	3 (2%)	1 (1%)	2 (7%)	n.s.
Chemotherapy				
(Neo-)adjuvant	40 (27%)	31 (26%)	9 (31%)	
Adjuvant	18 (12%)	13 (11%)	5 (17%)	n.s.
Radiotherapy				
(Neo-)adjuvant	11 (7%)	10 (8%)	1 (3%)	
Adjuvant	8 (5%)	7 (6%)	1 (3%)	n.s.
Local recurrence	17 (11%)	16 (13%)	1 (3%)	n.s.

**Table 2 cancers-13-05125-t002:** Demographic data, tumor characteristics and anatomic amputation levels. Age (range in brackets), else percentage in brackets.

	Total (*n* = 149)	Group I (*n* = 120)	Group II (*n* = 29)	*p* Value
Median age (years)	58 (13–89)	58 (13–89)	53 (17–79)	n.s.
Histological subtype				
Osteosarcoma	35 (24%)	24 (20%)	11 (38%)	
Chondrosarcoma	18 (12%)	17 (14%)	1 (3%)	
Undifferentiated sarcoma	17 (11%)	12 (19%)	5 (17%)	
Synovial sarcoma	11 (7%)	8 (7%)	3 (10%)	
Malignant fibrous histiocytoma	8 (5%)	7 (6%)	1 (3%)	
Leiomyosarcoma	7 (5%)	4 (3%)	3 (10%)	
Myxofibrosarcoma	7 (5%)	7 (6%)	0 (0%)	
Liposarcoma	6 (4%)	6 (5%)	0 (0%)	
Others	40 (27%)	35 (29%)	5 (17%)	n.s.
Grade (if applicable and recorded)				
I	4 (4%)	4 (5%)	0 (0%)	
II	32 (33%)	27 (35%)	5 (26%)	
III	61 (63%)	47 (60%)	14 (74%)	n.s.
Size				
<5 cm	20 (19%)	20 (23%)	0 (0%)	
5–10 cm	38 (35%)	31 (35%)	7 (35%)	
>10 cm	50 (46%)	37 (42%)	13 (65%)	*p* = 0.04
Site				
Upper extremity	39 (26%)	35 (29%)	4 (14%)	
Lower extremity	103 (69%)	78 (65%)	25 (86%)	
Pelvis	7 (5%)	7 (6%)	0 (0%)	n.s.
Type of amputation				
Transfemoral	50 (34%)	34 (28%)	16 (55%)	
Transtibial	25 (17%)	21 (18%)	4 (14%)	
Exarticulation hip	13 (9%)	10 (8%)	3 (10%)	
Transhumeral	13 (9%)	9 (8%)	4 (14%)	
Lower arm	8 (5%)	8 (7%)	0 (0%)	
Pelvis	8 (5%)	8 (7%)	0 (0%)	
Exarticulation knee	8 (5%)	7 (6%)	1 (3%)	
Exarticulation shoulder	8 (5%)	7 (6%)	1 (3%)	
Interscapulothoracic	7 (5%)	7 (6%)	0 (0%)	
Partial foot	5 (3%)	5 (4%)	0 (0%)	
Partial hand	4 (3%)	4 (3%)	0 (0%)	n.s.

## Data Availability

The datasets used and/or analyzed during the current study are available from the corresponding author on reasonable request.

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
