# Peer review of "Amputation for Extremity Sarcoma: Indications and Outcomes"

_cancers, 2021, doi:10.3390/cancers13205125_

Round 1

Reviewer 1 Report

Amputation for Extremity Sarcoma: Indications and Outcomes

Aim of this study is to answer the question, whether patients who undergo primary amputation (n=120) might have a better prognosis than those who require an amputation because of local recurrence or for contaminated margins (n=29) after limb sparing surgery. Based on the results the authors stated that both groups showed the same worse oncological results with respect to overall survival, probably due the high numbers of metastatic patients and that amputation is still a valid option in the treatment. The high  numbers of metastatic disease also reflect the selection bias of this group of patients.

The study is well written and easy to read. However there are some major concerns.

Major concerns

In this study both soft tissue sarcoma and bone sarcoma (osteosarcoma and chondrosarcoma patients; and probably some Ewing sarcoma patients in the group others) were taken together in one cohort. Also patients with sarcoma of the pelvis are included in this analysis and besides adult patients also pediatric patients are included in this cohort.

Since both the treatment and the prognosis of patients with soft tissue sarcoma and bone sarcoma differ considerably, and that of pediatric patients and adult patients too, especially for the soft tissue sarcoma, my question is to what extent conclusions can be drawn based on this already small patient population. Also 7 patients with a tumor of the pelvis were included, all included in the primary amputation group. In addition, histology of the tumor makes a huge difference in treatment and outcome. Excluding these patients from this study seems wise as the data from this small population are already far from robust.

Please clarify clearly, based on literature, why taking these different subgroups together is justified (soft tissue vs bone; pediatric vs adult patients and extremity vs pelvic). The introduction and also discussion is rather short, and the references used are mainly based on the knowledge of soft tissue sarcomas.

Please remove the data of the patients with pelvic sarcoma, or give more detailed information about kind of sarcoma and whether the groups are similar.

Minor concerns:

Introduction

At the end of the introduction the authors wrote that they would like to compare the patient who underwent an amputation due to local complications compared to those who require an amputation because of local recurrence or for contaminated margins after a LSS. I find the term local complications rather than primary amputation in this sentence rather confusing. Please explain in the text more clearly the reasons for primary resection as shown in Table 1 and use also the word primary amputation in this sentence.

Discussion

  • Line 143 “In general, patients with a need for amputation do have a worse prognosis since they typically have larger tumors, involvement of critical structures or multi compartmental local recurrences.” Please add some literature references, both for soft tissue sarcoma as bone sarcomas.
  • Between Line 161 and 165 there is an illogical transition: ”Our hypothesis that those patients who had an amputation due to local complications might have a better prognosis than those with LR or contaminated margins could not be proven on the basis of statistical significance. However, a trend towards such a difference was apparent and with only 29 patients in group II (versus 120 in group I), this could potentially also be caused by a lack of statistical power.

Maybe better to change line 165 in: “However, our finding that patients with primary and secondary amputations did have the same prognosis (Fig. 1), is identical to the results published by Stevenson et al.

  • In the summary it is stated that amputation is still a valid option in treating sarcoma patients. Please clarify more clearly in the discussion why are these results are so important for clinical practice.
  • In the discussion the influence of multimodality therapy and outcome (RT/chemotherapy) is not discussed. Since this is not unimportant, please add some literature (e.g recent article Roland et al.), and please make a correlation with your own results.

Author Response

Dear Editors,

Thank you very much for reviewing this manuscript. We think that the comments of the reviewers will increase the clarity of this study.

As proposed by the reviewers we made the following corrections:

Reviewer #1:

  1. In this study both soft tissue sarcoma and bone sarcoma (osteosarcoma and chondrosarcoma patients; and probably some Ewing sarcoma patients in the group others) were taken together in one cohort. Also patients with sarcoma of the pelvis are included in this analysis and besides adult patients also pediatric patients are included in this cohort. Since both the treatment and the prognosis of patients with soft tissue sarcoma and bone sarcoma differ considerably, and that of pediatric patients and adult patients too, especially for the soft tissue sarcoma, my question is to what extent conclusions can be drawn based on this already small patient population. Also 7 patients with a tumor of the pelvis were included, all included in the primary amputation group. In addition, histology of the tumor makes a huge difference in treatment and outcome. Excluding these patients from this study seems wise as the data from this small population are already far from robust. Please clarify clearly, based on literature, why taking these different subgroups together is justified (soft tissue vs bone; pediatric vs adult patients and extremity vs pelvic). The introduction and also discussion is rather short, and the references used are mainly based on the knowledge of soft tissue sarcomas. Please remove the data of the patients with pelvic sarcoma, or give more detailed information about kind of sarcoma and whether the groups are similar.

In general this comments are rational and well considered. Since years we ask the scientific community to seperate the entities and the locations because we know that both factors do influence treatment and prognosis of the patients. But if we would do that, even in a large cohort of patients as ours, at the end such a small number of patients in the subgroups would result, that no conclusions at all could be drawn. We had a look to that issue in literature. Papakonstantinou et al. published 2020 a metaanalysis of osteosarcoma patients only treated either by LS or amputation. The numbers of amputated patients in those studies there: 53, 27, 38, 40, 42, 36, 15, 300, 15, 95, 48, 46 and 143. In total 9/13 studies had a number below 50 patients. The studies with larger numbers as 143 or 300 are out of nationwide cohorts as SEER or the Japanese register.[1]

Those register studies of course allow large numbers in precisely defined subgroups as pelvic chondrosarcoma patients with a profound matching of 131 patients in each of two groups (amputated vs. LS, National Cancers Database, USA) but they share all the disadvantages of retrospective nationwide databases.[2] More than those national registers metaanalyses of data as for osteosarcoma only (all age) (934 LS vs. 662 amputated) might attract with high numbers but difficulties in comparing the single studies.[3]

Single center studies do have problems in reaching sufficient numbers. A large study published from Brigham and Women’s and Dana-Faber in 2018 had 54 extremity sarcoma (including “buttock”) patients of mixed bone and soft tissues sarcomas in 10 years.[4] There are some studies including only subgroups as soft tissue sarcomas but they ended with small numbers as 18 (Vanderbilt University Medical center)[5] or 39 (University Medical Center Groningen)[6]. Even mixed groups of bone and soft tissue sarcomas from recent years reached sometimes only small numbers as 24 (Jacksonville, Gainesville, FL, USA)[7]. If the authors try to focus on location as distal tibia and entity as osteosarcoma the resulting numbers are as small as 19 amputations even in a large center as the Rizzoli[8] or 25 patients with soft tissue sarcoma of the extremity at Mount Sinai Hospital, Toronto.[9]

Very few studies end with sufficient monocentric numbers in clearly defined subgroups as the 2015 published study from Birmingham comparing 197 patients with LS to 127 amputated patients in extremity osteosarcomas only.[10] Also Rizzoli published their osteosarcoma only data (location “limb”) with 95 amputations in 2002.[11]

Regarding the effect of combining children and adults we agree in general with the reviewer. Having a closer look in our data we saw 7 mainly older children (13,14,15,15,16, 16,17 years) all with bone sarcomas. So we think this does not affect the results.

So how to proceed with the in total justified critic of the reviewer? We hope that the reviewer will accept, that we take his critic to the discussion section and to the section “Limitations of the study”. We also changed the introduction section.

We included the following:

Introduction

“There are studies in osteosarcoma patients which describe a better local control with amputation but no survival benefit over LSS in patients with intralesional or marginal margins[10] but also two metaanalyses showing higher 5-year survival rates for LSS.[1] Regarding soft tissue sarcomas no difference in overall survival could be shown in two studies.[5] Regarding primary or secondary amputations in localized extremity sarcoma no difference in oncological outcome was published by Erstad et al.  in 2018.[4,12]”

Patients and Methods

“Only 7 children (13 - 17 years) all with bone sarcomas had been included.”

Discussion

„In this study patients with bone and soft tissue sarcomas including also 8 patients who needed an amputation at the level of the pelvis are included. A separation of entities and locations may have advantages because we know that both factors do influence treatment and prognosis of the patients. But at the end such a small number of patients in the subgroups would result, that drawing any conclusions would be difficult. We had a look to that issue in literature. Papakonstantinou et al. published 2020 a metaanalysis of osteosarcoma patients only treated either by LSS or amputation. The numbers of amputated patients in those studies there: 53, 27, 38, 40, 42, 36, 15, 300, 15, 95, 48, 46 and 143. In total 9/13 studies had a number below 50 patients. The studies with larger numbers as 143 or 300 are out of nationwide cohorts as SEER or the Japanese register.[1]. Those register studies of course allow large numbers in precisely defined subgroups as pelvic chondrosarcoma patients with a profound matching of 131 patients in each of two groups (amputated vs. LSS, National Cancers Database, USA)[2] but they share all the disadvantages of retrospective nationwide databases. More than those national registers metaanalyses of data as for osteosarcoma only (all age) (934 LSS vs. 662 amputated) might attract with high numbers but difficulties in comparing the single studies.[3] Single center studies do have problems in reaching sufficient numbers. A large study of amputated patients published from Brigham and Women’s and Dana-Faber in 2018 had 54 extremity sarcoma (including “buttock”) patients of mixed bone and soft tissues sarcomas in 10 years.[4] There are some studies including only subgroups as soft tissue sarcomas but they ended with small numbers as 18[5] or 39[6]. Even mixed groups of bone and soft tissue sarcomas from recent years reached sometimes only small numbers as 24.[7] If the authors try to focus on location as distal tibia and entity as osteosarcoma the resulting numbers are as small as 19 amputations even in a large center as the Rizzoli[8] or 25 patients with soft tissue sarcoma of the extremity at Mount Sinai Hospital, Toronto.[9] Very few studies end with sufficient monocentric numbers in clearly defined subgroups as the 2015 published study from Birmingham comparing 197 patients with LSS to 127 amputated patients in extremity osteosarcomas only.[10] Also Rizzoli published their osteosarcoma only data (location “limb”) with 95 amputations in 2002.[11]“

Limitations ft he study

We included the sentences „The study cohort consists of bone and soft tissue sarcoma patients in different locations. A separation of entities and locations may have advantages but the general aspects of surgical sarcoma therapy apply to all.“

  1. Introduction: At the end of the introduction the authors wrote that they would like to compare the patient who underwent an amputation due to local complications compared to those who require an amputation because of local recurrence or for contaminated margins after a LSS. I find the term local complications rather than primary amputation in this sentence rather confusing. Please explain in the text more clearly the reasons for primary resection as shown in Table 1 and use also the word primary amputation in this sentence.

We focused on the term „secondary amputation“ as for example in the simple summary „Two groups with primary amputations (n=120) or secondary amputations after failed LSS due to LR or complications (n=29) were compared.“ There it was just added. In the remaining text we stick to these. We explained the indications for amputation on page 9 with respect to table 2 in the old text as follows:

„The indications for amputation for both groups are listed in Table 2. In group I multicompartmental involvement and the size of the tumor were the main reasons for amputation. In group II infection or ischemia after LSS were the major factors leading to amputation. Local recurrence (LR) as cause of a secondary amputation was seen in 14% of this group.“

We added now the sentence:

„Despite multicompartimental involvement LSS is possible in many cases. But in those patients amputated for that indication in this study reconstruction of the bone, the vessels, the nerves or the soft tissues in total sums up to an individually not acceptable surgical risk or functional disadvantage in respect to amputation.“

  1. Discussion: Line 143 “In general, patients with a need for amputation do have a worse prognosis since they typically have larger tumors, involvement of critical structures or multi compartmental local recurrences.” Please add some literature references, both for soft tissue sarcoma as bone sarcomas.

We added just as citation: M.A. Clark and J.M. Thomas: Amputation for soft-tissue sarcoma[13]: „However, generally the larger, high-grade, recurrent tumours are the ones that require amputation; predictably, overall prognosis is poor in this group.“, Ghert at al.: The indications for and the prognostic significance of amputation as the primary surgical procedure for localized soft tissue sarcoma of the extremity[9] : “ Twenty-five (6%) of 413 patients with STS underwent primary amputation: they were older (P = .05), had larger tumors (P = .001), and had a significantly greater risk of developing metastatic disease than patients who underwent limb-sparing procedures (P = .008).“

And for the bone sarcomas: Reddy et al.: Does amputation offer any survival benefit over limb salvage in osteosarcoma patients with poor chemonecrosis and close margins?[10]: „The mean size (maximum single dimension) of the tumour in the amputation group was 12.6 cm (8 to 42), compared with 12.2 cm (6 to 27) in the LSS (intralesional) group and 10.8 cm (6 to 22) in the LSS (marginal margins) group. This was statistically significant (p = 0.02).“

We did not include more literature showing the general risk factors for sarcomas on prognosis. We think we reader will know them.

  1. Discussion: Between Line 161 and 165 there is an illogical transition: ”Our hypothesis that those patients who had an amputation due to local complications might have a better prognosis than those with LR or contaminated margins could not be proven on the basis of statistical significance. However, a trend towards such a difference was apparent and with only 29 patients in group II (versus 120 in group I), this could potentially also be caused by a lack of statistical power. Maybe better to change line 165 in: “However, our finding that patients with primary and secondary amputations did have the same prognosis (Fig. 1), is identical to the results published by Stevenson et al.

We reviewer is right, we should change that. But our secondary amputations consists of two groups, the tumor related (margins, LR) and the complication related (infection).

Stevenson compared just primary and secondary amputations. He did not separate the secondary amputations into whose with tumor related indication and complication related issues.

To make that mor clear we changed the paragraph to:

„Our hypothesis that those patients who had the secondary amputation due to local complications (and not a tumor related issue) might have a better prognosis than those with a secondary amputation due to LR or contaminated margins could not be proven on the basis of statistical significance. However, a trend towards such a difference was apparent and with only 29 patients in group II (versus 120 in group I), this could potentially also be caused by a lack of statistical power.“

  1. Discussion: In the discussion the influence of multimodality therapy and outcome (RT/chemotherapy) is not discussed. Since this is not unimportant, please add some literature (e.g recent article Roland et al.), and please make a correlation with your own results.

In general the reviewer is right. More important to us is also the question whether multimodal therapy increases the risk of amputation (e.g. in neoadjuvant radiotherapy). This is part of another study comparing radiotherapy (and also chemotherapy) before and after surgery in our patients. For this study with such an inhomgenous patient cohort as discussed above we think that would neither show any significance nor help the reader. So we did not include that in the discussion and would be happy to stick to that.

  1. Summary: In the summary it is stated that amputation is still a valid option in treating sarcoma patients. Please clarify more clearly in the discussion why are these results are so important for clinical practice.

This is a good point. We added the sentence: „For clinical practice a secondary amputation after failed LSS does therefore not influence the oncological outcome of the patient but might influence the amputation level.“

Reviewer #2:

  1. I appreciate the paper. Anyway, given the relatively small number of patients involved considering also the long period of time observed, I feel that Your research doesn't reach enough priority to be published in Cancers.

As it is the number of patients published in other studies are much smaller than in this study. And this is a monocentric study with a comparative homogenous spectrum of indications and therapy. The reviewer is right, as more patients in the study as better but still we are happy to have such a number in this study. We think, we reader will benefit from this study.

  1. The bibliographic list is absolutely lacking and inadequate.

We enlarged the bibliography as stated and explained above now to 24 citations (before 15).

Editor

Please provide the institutional e-mail address of Dr. Maya Kirilova.

Miss Kirilova is a student. She did this study for her thesis. For this reason she does not have an institutional e-mail address and uses: m.hkirilova@gmail.com

Please provide the full name of LMU.

Ludwig-Maximilians-University was added for LMU

Please add the highlight content (post code, city, country).

We added D-81377 Munich, Germany

Words in main text should > 3000, please add more words.

As stated above, content was added. Now we count 2.973. We are happy to have a short text, but if necessary we could enlarge it.

Please revised the format of reference as [1-5]

Done.

We compared your paper with the publications ,and find there are some overlaps. Please re-write the highlighted parts in your own words.

Yes, we used the same wording in our last publication.

„Prior to surgery, magnetic resonance imaging (MRI) and in select cases computed tomography (CT) were used to define size and localization of the tumor. A CT scan of the chest in early years also a radiograph was performed to determine the presence or absence of pulmonary metastatic disease, which was repeated at follow-up. The margin status of all resections was evaluated.“

Was changed to:

„For local staging, magnetic resonance imaging (MRI) and in some cases computed tomography (CT) were used to clarify the location of the tumor and the extension in respect to vital structures as vessels or nerves or neighboring compartiments. A CT scan of the chest or in early years a thoraxic radiograph was used for diagnosing pulmonary metastatic disease. This and also local MRI was repeated for follow-up. In all resections the margin status was evaluated by using the method of applying ink to the specimen.“

And:

„In this retrospective evaluation, the patients were analyzed with regards to local re-currence and metastatic disease with the main end points being LRFS and OS. All patients were followed for evidence of LR or distant metastasis. LRFS and OS were defined either as the time from amputation to the first occurrence of LR or to death from any cause. For statistical analysis, OS and LRFS were calculated according to the Kaplan-Meier method. Significance analysis was performed using the Log-Rank or the Cox Proportional-Hazards Regression model. A p value of less than 0.05 was considered statistically significant. The data analysis software used was MedCalc® (MedCalc Software, Ostend, Belgium).“

Was changed to:

„In this retrospective study, follow-up of the patients was analyzed in respect to local recurrence (LR) and metastatic disease with the main end points being LRFS and OS. All patients were followed for evidence of LR or distant metastasis as described above. LRFS and OS were defined either as the time from amputation to the first occurrence of LR or to death from any cause. For statistical analysis, OS and LRFS were calculated according to the Kaplan-Meier method. Significance analysis was performed using the Log-Rank or the Cox Proportional-Hazards Regression model. A p value of less than 0.05 was considered statistically significant. The data analysis software used was MedCalc® (MedCalc Software, Ostend, Belgium).“

Please provide the ethic code number.

„This study was approved by the ethics committee of the Medical Faculty, University of Munich. Written consent was obtained from all the surviving patients included in this study.“

Was changed to

Institutional Review Board Statement:

„This study was approved by the ethics committee of the Medical Faculty, University of Munich (17-891).“

Informed Consent Statement: Please add “Informed consent was obtained from all subjects in-volved in the study.” OR “Patient consent was waived due to REASON (please provide a detailed justification).” OR “Not applicable” for studies not involving humans.

We added:

„Informed Consent Statement: Informed consent was obtained from all surviving patients included in this study. For non-surviving patients data were irreversibly anonymized as recommended by the ethics committee.“

Please add, and please provide the blank copy of written informed consent.

We uploaded a blank copy.

We added percentage signs „%“ in the tables to the responding numbers.

Thank you once again for reviewing this manuscript!

For the authors,

Sincerely yours,

Prof. Dr. H.R. Dürr

  1. Papakonstantinou, E.; Stamatopoulos, A.; D, I.A.; Kenanidis, E.; Potoupnis, M.; Haidich, A.B.; Tsiridis, E. Limb-salvage surgery offers better five-year survival rate than amputation in patients with limb osteosarcoma treated with neoadjuvant chemotherapy. A systematic review and meta-analysis. J Bone Oncol 2020, 25, 100319, doi:10.1016/j.jbo.2020.100319.
  2. Kim, C.Y.; Collier, C.D.; Liu, R.W.; Getty, P.J. Are Limb-sparing Surgical Resections Comparable to Amputation for Patients With Pelvic Chondrosarcoma? A Case-control, Propensity Score-matched Analysis of the National Cancer Database. Clinical orthopaedics and related research 2019, 477, 596-605, doi:10.1097/corr.0000000000000622.
  3. Han, G.; Bi, W.Z.; Xu, M.; Jia, J.P.; Wang, Y. Amputation Versus Limb-Salvage Surgery in Patients with Osteosarcoma: A Meta-analysis. World journal of surgery 2016, 40, 2016-2027, doi:10.1007/s00268-016-3500-7.
  4. Erstad, D.J.; Ready, J.; Abraham, J.; Ferrone, M.L.; Bertagnolli, M.M.; Baldini, E.H.; Raut, C.P. Amputation for Extremity Sarcoma: Contemporary Indications and Outcomes. Annals of surgical oncology 2018, 25, 394-403, doi:10.1245/s10434-017-6240-5.
  5. Alamanda, V.K.; Crosby, S.N.; Archer, K.R.; Song, Y.; Schwartz, H.S.; Holt, G.E. Amputation for extremity soft tissue sarcoma does not increase overall survival: a retrospective cohort study. European journal of surgical oncology : the journal of the European Society of Surgical Oncology and the British Association of Surgical Oncology 2012, 38, 1178-1183, doi:10.1016/j.ejso.2012.08.024.
  6. Stevenson, M.G.; Musters, A.H.; Geertzen, J.H.B.; van Leeuwen, B.L.; Hoekstra, H.J.; Been, L.B. Amputations for extremity soft tissue sarcoma in an era of limb salvage treatment: Local control and survival. Journal of surgical oncology 2018, 117, 434-442, doi:10.1002/jso.24881.
  7. Wilke, B.; Cooper, A.; Scarborough, M.; Gibbs, P.; Spiguel, A. A Comparison of Limb Salvage Versus Amputation for Nonmetastatic Sarcomas Using Patient-reported Outcomes Measurement Information System Outcomes. The Journal of the American Academy of Orthopaedic Surgeons 2019, 27, e381-e389, doi:10.5435/jaaos-d-17-00758.
  8. Mavrogenis, A.F.; Abati, C.N.; Romagnoli, C.; Ruggieri, P. Similar survival but better function for patients after limb salvage versus amputation for distal tibia osteosarcoma. Clinical orthopaedics and related research 2012, 470, 1735-1748, doi:10.1007/s11999-011-2238-7.
  9. Ghert, M.A.; Abudu, A.; Driver, N.; Davis, A.M.; Griffin, A.M.; Pearce, D.; White, L.; O'Sullivan, B.; Catton, C.N.; Bell, R.S.; et al. The indications for and the prognostic significance of amputation as the primary surgical procedure for localized soft tissue sarcoma of the extremity. Annals of surgical oncology 2005, 12, 10-17, doi:10.1007/s10434-004-1171-3.
  10. Reddy, K.I.; Wafa, H.; Gaston, C.L.; Grimer, R.J.; Abudu, A.T.; Jeys, L.M.; Carter, S.R.; Tillman, R.M. Does amputation offer any survival benefit over limb salvage in osteosarcoma patients with poor chemonecrosis and close margins? The bone & joint journal 2015, 97-b, 115-120, doi:10.1302/0301-620x.97b1.33924.
  11. Bacci, G.; Ferrari, S.; Lari, S.; Mercuri, M.; Donati, D.; Longhi, A.; Forni, C.; Bertoni, F.; Versari, M.; Pignotti, E. Osteosarcoma of the limb. Amputation or limb salvage in patients treated by neoadjuvant chemotherapy. The Journal of bone and joint surgery. British volume 2002, 84, 88-92, doi:10.1302/0301-620x.84b1.12211.
  12. Erstad, D.J. ASO Author Reflections: Amputation for Extremity Sarcoma. Annals of surgical oncology 2019, 26, 548, doi:10.1245/s10434-018-7069-2.
  13. Clark, M.A.; Thomas, J.M. Amputation for soft-tissue sarcoma. The Lancet. Oncology 2003, 4, 335-342, doi:10.1016/s1470-2045(03)01113-6.

Reviewer 2 Report

I appreciate the paper.
Anyway, given the relatively small number of patients involved considering also the long period of time observed, I feel that Your research doesn't reach enough priority to be published in Cancers.

The bibliographic list is absolutely lacking and inadequate.

Author Response

(The authors gave the same response as above.)

Round 2

Reviewer 1 Report

The authors have well refuted earlier points and elaborated them satisfactorily. Except that the study cohort is still very small and diverse, leaving doubts about the extrapolability of the data for all subgroups. 

Author Response

The thank the reviewer again for his help in the first round which has significantly improved the manuscript. As described we think any further subgroup analysis will have a bias and results in inappropriate patient numbers.

Reviewer 2 Report

The paper has improved after revisions.

The criticalities I observed (very long interval time and relatively small samples compared  time) are underlined in limitations and seems tolerable, so i suggest some concerns to improve the paper:

  • Please discuss briefly about the other factors that lead to the amputation that can be considered themselves  the causes of poor prognosis (e.g. vascular proximity/infiltration, larger sizes, general signs of infiltration...). Also, you didn t perform a multivariate analyses considering all prognostic factors. Add this to the limits and discuss it briefly.
  • More generally, highlight better (maybe with a brief paragraph) discussing the reasons that usually lead to amputation in soft tissue and bone sarcomas.
  • The quality of paper's presentation would benefit from the addition of some Images (e.g. a couple of MRI and surgical pictures to show the reason that lead to amputation)

Thank You

Author Response

The paper has improved after revisions. The criticalities I observed (very long interval time and relatively small samples compared  time) are underlined in limitations and seems tolerable, so I suggest some concerns to improve the paper:

  1. Please discuss briefly about the other factors that lead to the amputation that can be considered themselves the causes of poor prognosis (e.g. vascular proximity/infiltration, larger sizes, general signs of infiltration...).

We wrote in the manuscript in the results section:

“Despite multicompartimental involvement LSS is possible in many cases. But in those patients amputated for that indication in this study reconstruction of the bone, the vessels, the nerves or the soft tissues in total sums up to an individually not acceptable surgical risk or functional disadvantage in respect to amputation.”

In the discussion (2. sentence) we now added a new paragraph:

“As stated above amputation for oncological reasons may be considered a bias in respect to worse oncological outcome. Vascular infiltration is a known worse prognostic factor in osteosarcoma as also bone invasion in soft tissue sarcoma.[1-3]The involvement of neurovascular structures in comparison has either no or a less significant influence on prognosis.[2,3] Also larger size, which in many cases together with the infiltration of neurovascular structures predicts amputation, is a well established single worse prognostic factor.[4]”

  1. Also, you didn t perform a multivariate analyses considering all prognostic factors. Add this to the limits and discuss it briefly.

We added the following two sentences to the Limitations of the study section after the sentence “A separation of entities and locations may have advantages but the general aspects of surgical sarcoma therapy apply to all.” closing this paragraph:

“We are well aware that this study does not investigate or consider the known prognostic factors in sarcoma patients. This study cohort of amputees is highly selected in respect to worse prognostic factors in the group of patients amputated for oncological reasons.”

  1. More generally, highlight better (maybe with a brief paragraph) discussing the reasons that usually lead to amputation in soft tissue and bone sarcomas.

In the Introduction we stated:

“Despite the advances in LSS including free vascular flaps or extended neurovascular resections and reconstructions, amputation is still a valid option. If limb function is insufficient, local recurrence (LR) with widespread contamination leaves no other option or infection and/or ischemia after LSS could not be treated otherwise, amputation is still indicated. [8] In the rare cases with exulcerating, fungating tumors, amputation might be the most appropriate palliative procedure.”

And also in the Results section we wrote the paragraph:

“Despite multicompartimental involvement LSS is possible in many cases. But in those patients amputated for that indication in this study reconstruction of the bone, the vessels, the nerves or the soft tissues in total sums up to an individually not acceptable surgical risk or functional disadvantage in respect to amputation.”

Together with the new sentences added to the discussion we think, this issue is hopefully discussed in a sufficient manner. We are well aware, that this manuscript does not give a guideline to amputation in sarcoma surgery but provides information to the reader what could be expected in respect to prognosis in this group of patients.

  1. The quality of paper's presentation would benefit from the addition of some Images (e.g. a couple of MRI and surgical pictures to show the reason that lead to amputation).

This is an important issue. Thank you very much! We included 3 new figures and renumbered the others:

FIGURE 1: Axial MRI of an Undifferentiated Pleomorphic Sarcoma of the right lower leg infiltrating the bone and the major vessels and nerves.

FIGURE 2: Radiographs and axial MRI of a patient with an osteosarcoma. After pathological fracture an osteosynthesis has induced widespread contamination and tumor growth of the whole distal upper calf.

FIGURE 3: More than 30 years after treatment of an Ewing sarcoma of the humerus with local radiation and chemotherapy complications as chronic osteomyelitis, nerve palsy and edema indicated an amputation due to functional reasons.

Round 3

Reviewer 2 Report

I am satisfied with the revisions performed.

Author Response

We thank the reviewer for his help with this manuscript!